

# Analysis of the apoplast fluid proteome during the induction of systemic acquired resistance in *Arabidopsis thaliana*

Shuna Jiang[1], Liying Pan[2], Qingfeng Zhou[2], Wenjie Xu[2], Fuge He[2], Lei Zhang[3] and Hang Gao[2]

[1] College of Survey and Planning, Shangqiu Normal University, Shangqiu, China
[2] College of Biology and Food, Shangqiu Normal University, Shangqiu, China
[3] Institute of Crops Molecular Breeding, Henan Academy of Agricultural Sciences, Zhengzhou, China

## ABSTRACT

**Background:** Plant-pathogen interactions occur in the apoplast comprising the cell wall matrix and the fluid in the extracellular space outside the plasma membrane. However, little is known regarding the contribution of the apoplastic proteome to systemic acquired resistance (SAR).

**Methods:** Specifically, SAR was induced by inoculating plants with *Pst* DC3000 avrRps4. The apoplast washing fluid (AWF) was collected from the systemic leaves of the SAR-induced or mock-treated plants. A label free quantitative proteomic analysis was performed to identified the proteins related to SAR in AWF.

**Results:** A total of 117 proteins were designated as differentially accumulated proteins (DAPs), including numerous pathogenesis-related proteins, kinases, glycosyl hydrolases, and redox-related proteins. Functional enrichment analyses shown that these DAPs were mainly enriched in carbohydrate metabolic process, cell wall organization, hydrogen peroxide catabolic process, and positive regulation of catalytic activity. Comparative analysis of proteome data indicated that these DAPs were selectively enriched in the apoplast during the induction of SAR.

**Conclusions:** The findings of this study indicate the apoplastic proteome is involved in SAR. The data presented herein may be useful for future investigations on the molecular mechanism mediating the establishment of SAR.

## INTRODUCTION

Plants have evolved different defense systems to resist attacks by pathogens, including both innate and inducible immune systems (*Spoel & Dong, 2012*; *Shah & Zeier, 2013*). Plant innate immunity, which occurs in the pathogen-infected tissues, can be broadly divided into two different layers, namely pathogen-associated molecular patterns (PAMP)-triggered immunity (PTI) and effector-triggered immunity (ETI). In response to an infection by a microbial pathogen, PAMPs are recognized by pattern recognition receptors (PRRs) localized on the surface of plant cells, resulting in the activation of PTI (*Gust, Pruitt*

Corresponding authors
Lei Zhang, zhanglei7971@163.com
Hang Gao, gaohangsqsy@163.com

& *Nurnberger, 2017*; *Naveed et al., 2020*). The successful activation of PTI leads to an oxidative burst, the reinforcement of the cell wall (*i.e.*, increased rigidity), and the synthesis of antimicrobial compounds and pathogenesis-related (PR) proteins (*Newman et al., 2013*). During infections, many pathogens evade the effects of PTI-related mechanisms by releasing effectors into host cells. Nevertheless, most plants can recognize these effectors through intracellular nucleotide-binding domain leucine-rich repeat containing receptors (NLRs) and activate more robust immune responses, known as ETI (*Naveed et al., 2020*).

Plant inducible immunity, which is associated with systemic tissues free of pathogens, can be classified as systemic acquired resistance (SAR) and induced systemic resistance (ISR), which are induced by pathogenic microbes and beneficial soil microbes, respectively (*Yu et al., 2022*). SAR is an inducible defense mechanism that is systemically activated in response to localized infection by a variety of pathogens or treatment with SAR inducers, which can induce strong and rapid immune responses to future infections by bacteria, fungi, viruses, and oomycetes in systemic tissues (*Gao et al., 2021*). To date, numerous putative SAR inducers have been identified. These include glycerol-3-phosphate (G3P) derivatives (*Chanda et al., 2011*), azelaic acid (AzA) (*Jung et al., 2009*), dehydroabietinal (DA) (*Chaturvedi et al., 2012*), N-hydroxy-pipecolic acid (NHP) (*Navarova et al., 2012*; *Hartmann et al., 2018*; *Chen et al., 2018*), nitric oxide (NO) and ROS (*Wang et al., 2014*; *El-Shetehy et al., 2015*), pinenes (*Riedlmeier et al., 2017*; *Wenig et al., 2019*), pyridine nucleotides (NADP) (*Wang et al., 2019*), salicylic acid (SA) (*Lim et al., 2020*; *Kachroo, Liu & Kachroo, 2020*), methyl salicylate (MeSA) (*Park et al., 2007*), β-ionone, and nonanal (*Brambilla et al., 2022*). Among these compounds, AZA, G3P, DA, SA and NHP can be transported to systemic tissues through the phloem, whereas MT, MeSA, pinenes, β-ionone and nonanal are transported *via* volatilization. The induction of SAR by all of these SAR inducers depends on the SA signaling pathway. NHP is a lysine derivative produced in reactions catalyzed by AGD2-LIKE DEFENSE RESPONSE PROTEIN1 (ALD1), SAR-DEFICIENT4 (SARD4), and FLAVINDEPENDENT MONOOXYGENASE1 (FMO1) in *Arabidopsis thaliana*. This compound is crucial for SAR in a variety of plant species (*Holmes et al., 2019*; *Zeier, 2021*; *Schnake et al., 2020*). Recent studies have revealed that *A. thaliana* UGT76B1 maintains normal plant growth and development by converting SA/NHP to the inactive SAG/NHPG (*Bauer et al., 2021*; *Cai et al., 2021*; *Mohnike et al., 2021*; *Pastorczyk-Szlenkier & Bednarek, 2021*).

The apoplast, including the cell wall matrix and the fluid in the extracellular space outside the plasma membrane, is an enclosed active battle field where pathogen-host interactions occurs (*Delaunois et al., 2014*). The plant cell wall, composed of cellulose, hemicellulose, pectin, and protein, acts as the primary physical barrier against pathogen colonization of plant cells. The increased cell wall integrity following a pathogen attack is related to the deposition of callose and the formation of papillae at infection sites (*Delaunois et al., 2014*; *Zhang et al., 2020*). Initially, disease resistance was simply attributed to increased cell wall rigidity (*Bacete et al., 2018*). However, numerous studies in recent years demonstrated that defense response-related signaling pathways are activated in plants that are defective in cell wall synthesis or modification (*Miedes et al., 2014*; *Nafisi, Fimognari & Sakuragi, 2015*; *Houston et al., 2016*), implying that the cell wall is also

involved in the transduction of defense-related signals. To successfully penetrate the plant cell wall, some pathogens secrete a variety of cell wall-degrading enzymes. The detection of the damage-associated molecular patterns (DAMPs) resulting from the degraded cell wall results in the activation of DAMP-triggered immunity (DTI), suggesting the cell wall is involved in a signal transduction pathway that leads to plant immunity (*Bacete et al., 2018*; *Zhang et al., 2020*; *De Lorenzo et al., 2019*). In many plant species, DAMPs are perceived by wall-associated kinase 1 (WAK1), which subsequently activates a wide range of defense responses, including ROS production, callose deposition, and phytoalexin and PR protein accumulation, as part of the DTI mechanism (*Ferrari et al., 2013*). The apoplastic fluid circulating in the intercellular spaces is crucial for intracellular communication. The abundant metabolites and proteins present in the apoplast are thought to play important roles in PTI and ETI, including signal perception and transduction, reactive oxygen species (ROS) accumulation, programmed cell death (PCD), and the secretion of defense-related proteins and metabolites (*Doehlemann & Hemetsberger, 2013*; *Guerra-Guimaraes et al., 2015*).

To date, the apoplastic proteome and metabolome in pathogen-infected plant tissues have been characterized by many works (*Martínez-González et al., 2018*). *Cheng et al. (2009)* investigated the secretory proteome of suspension-cultured *Arabidopsis* cells in response to SA treatment, and found 63 secreted proteins were induced by SA. In maize, the apoplast proteins responsive to *Fusarium verticillioides* are related to signal transduction, cell wall modifications, carbohydrate metabolism, and cell redox homeostasis (*Hafiz et al., 2022*). *Guerra-Guimaraes et al. (2015)* investigated the apoplastic proteome of coffee plants infected with *Hemileia vastatrix*, and identified numerous resistance-related proteins, including pathogen related-like proteins (PR-proteins), serine proteases, and glycohydrolases in the cell wall. Using metabolomic and ion analysis techniques, O'Leary and coworkers observed that citrate, γ-aminobutyrate (GABA), metal ions (*e.g.*, $K^+$, $Ca^{2+}$, $Fe^{2/3+}$, and $Mg^{2+}$), sucrose, β-cyanoalanine, and several amino acids enriched in the apoplast of *Phaseolus vulgaris* leaves infected with *Pseudomonas syringae* pv. *phaseolicola* (*O'Leary et al., 2016*). The SAR-related proteins in phloem exudates of *Pst* DC3000 inoculated leaves, which represent the mobile proteins in phloem loaded *via* apoplast and plasmodesmata, were identified using a label-free method (*Carella et al., 2016*). However, it remains unclear whether or how the apoplastic proteome of pathogen-free systemic leaves is affected by the induction of SAR. In this study, we performed comprehensive quantitative proteomic analyses to elucidate the SAR-induced changes in the apoplast components of uninfected systemic leaves. Our findings have implications for future investigations on the molecular mechanism underlying the induction of SAR.

## MATERIALS AND METHODS

### Plant materials and growth conditions

Wild-type Col-0 *A. thaliana* plants were sown in individual pots containing vermiculite, perlite, and nutrient soil (1:1:1, v/v/v), and grown in a growth chamber set at 22 °C with a

12-h light (photon flux density 70 µmol m$^{-2}$s$^{-1}$) /12-h dark cycle and 65% relative humidity. Four-week-old plants were used for the subsequent SAR analysis.

## Bacterial culture and inoculation

*Pseudomonas syringae* pv. tomato DC3000 (*Pst* DC3000) and *Pst* DC3000 expressing avrRps4 (Pst DC3000 avrRps4) were grown overnight at 28 °C in King's broth liquid medium. Bacteria were harvested by centrifugation (3,000 rpm, 5 min). After rinsing once with 10 mM MgCl$_2$, the bacteria were resuspended in 10 mM MgCl$_2$ (OD600 = 0.005) and then used to induce SAR. Briefly, three leaves from 28-day-old plants (typically leaves 4–6) were infiltrated with *Pst* DC3000 avrRps4 using a 1 mL syringe. After 2 days, six systemic leaves (typically leaves 7–12) for one plant were used for apoplastic washing fluid collection. To confirm SAR was induced, the systemic leaves were secondary inoculated with *Pst* DC3000, and a bacterial growth assay was completed as previously described (*Carella et al., 2016*). Six replicates were performed. Student' *t*-test was used to compare group differences.

## SAR experiments, and apoplastic washing fluid collection

The collected systemic leaves were used for the extraction of apoplastic washing fluid (AWF) as previously described (*Huang et al., 2021*). Briefly, fully expanded rosette leaves were cut at the base using a razor blade and then washed three times with distilled water to remove cytoplasmic contaminants from the damaged cells. The leaves were placed in a 200 mL needleless syringe and gently infiltrated with infiltration buffer by negative pressure (2 mM CaCl$_2$, 0.1 M NaCl, and 20 mM L2-(N-morpholino)ethanesulfonic acid hydrate, pH 6.0) for 10 s. This step was repeated 2–3 times until all detached leaves were infiltrated. The successfully infiltrated leaves are darker in color and translucent. The leaf surface was gently patted dry with a clean paper towel to eliminate residual infiltration buffer. The infiltrated leaves were carefully layered onto the sticky side of Scotch tape and then fixed onto a 15 mL centrifuge tube, which was placed in a bigger conical tube (50 mL) and centrifuged at 900 × g for 20 min at 4 °C to collect the AWF. After fractionating the AWF *via* successive centrifugations at 2,000 × g for 15 min, 5,000 × g for 20 min, and 12,000 × g for 30 min, it was filtered using a cellulose syringe filter (0.45 µm pore size). Typically, approximately 2 mL AWF was obtained from 120 *A. thaliana* plants (720 leaves) for each treatment, and was used as one biological replicate. The AWF samples were subsequently concentrated to approximately 200 µL using a centrifugal concentrator with a 3-kDa cutoff. The concentrated AWF solutions were stored at −80 °C. Three biological replicates for each treatment were used for the proteomic analysis.

## Electrolyte leakage and malate dehydrogenase assays

The integrity of the plasma membrane after the extraction of AWF was assessed by measuring the electric conductivity as described by *Campos et al. (2003)* and *Regente et al. (2017)*. Briefly, 1 g detached fresh leaves or leaves after the AWF extraction were washed three times with double-distilled water and then immersed in double-distilled water at 25 °C for 2 h. The conductance of the solution was measured using a conductivity meter.

Total conductivity was determined using a similar sample incubated at 80 °C for 2 h. Three biological replicates were performed. Student's *t*-test were considered as having statistically significant differences.

Malate Dehydrogenase Assay Kit (Solarbio, Beijing, China) was used to measure the activity of MDH, following the manufacturer's instructions to evaluate the cytoplasmic contamination in AWF. Three biological replicates were performed. Student's *t*-test were considered as having statistically significant differences.

## Real-time quantitative PCR (RT-qPCR) analysis

The successful induction of SAR was confirmed on the basis of the pathogenesis-related *PR1* expression level. Total RNA was isolated using the RNAiso Reagent (TaKaRa Bio, Otsu, Japan) and the RT-qPCR analysis was performed as described by *Zhou et al. (2021)*. *ACTIN8* (AT1G49240) was selected as the internal reference gene to calculate relative expression level fold-changes according to the comparative cycle threshold ($2^{-\Delta\Delta Ct}$) values. The following primers were used in this study: *ACTIN8* forward: TGTGCCTATCTACGAGGGTTT; *ACTIN8* reverse: TTTCCCGTTCTGCTGTTGT; *PR1* forward: GTGCTCTTGTTCTTCCCTCG; *PR1* reverse: GCCTGGTTGTGAACCCTTAG. Three biological replicates were performed. Student's *t*-test were considered as having statistically significant differences.

## Protein digestion and LC-MS/MS analysis

A previously published filter-aided sample preparation method was used for the digestion of the concentrated AWF proteins (*Wisniewski et al., 2009*). Briefly, proteins (200 μg) were washed three times with UA buffer (8 M urea and 150 mM Tris-HCl, pH 8.5) and then alkylated in 50 mM iodoacetic acid for 30 min at 25 °C in darkness. After washing three times with UA buffer, the proteins were digested with 2 μg trypsin (Promega, Madison, WI, USA) in 25 mM $NH_4HCO_3$ on a 30-kDa filter unit (Millipore, Burlington, MA, USA) for 18 h at 37 °C. The peptide concentration was determined based on the $OD_{280}$ value.

After the desalting step using a C18 cartridge, the peptides were analyzed using the Easy-nLC 1000 system coupled with the Orbitrap Fusion Lumos mass spectrometer (Thermo Fisher Scientific, Waltham, MA, USA). Total peptide solutions (1 μg from each sample) were separated using a peptide trap (Thermo Fisher Scientific, Waltham, MA, USA, EASY-Spray C18 column; 2 cm × 100 μm × 5 μm) and a 60 min linear solvent gradient (5–28% phase B (0.1% formic acid in 100% ACN), 5–40 min; 28–90% phase B, 40–42 min; 90% phase B, 42–60 min, buffer A (0.1% FA in $H_2O$)). The mass spectrometry (MS) operating parameters were as follows: data-dependent mode; MS1 resolution 60,000 at m/z 200; MS2 resolution 15,000 at m/z 120; m/z range 350–1,600 for the full scan.

## Data analysis and bioinformatics analysis

The raw data were together analyzed using MaxQuant (version 1.6.5.0) software. The MaxQuant searches were conducted with the 'match between runs' active. Tandem mass spectra were used to screen the *A. thaliana* TAIR 10 database. The false discovery rate threshold for the peptides and proteins was set to 1%. Peptides detected in all of the three

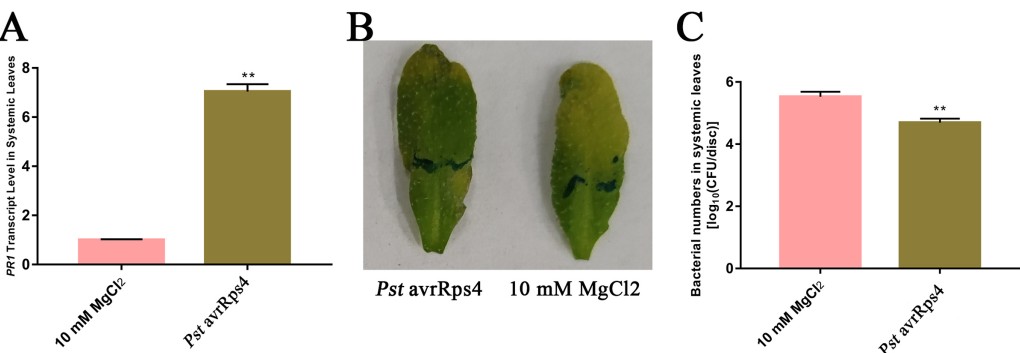

**Figure 1 Induction of SAR.** (A) *PR1* expression in systemic leaves of *Pst* DC3000 avrRps4 or 10 mM $MgCl_2$ locallyinoculated plants at 48 h. **$P < 0.01$ (Student's *t*-test), data are presented as the mean ± SE ($n = 3$). (B) Phenotypic changes of systemic leaves following secondary infection with *Pst* DC3000 at 3 days post-inoculation. (C) Bacterial growth analysis of *Pst* DC3000 in systemic leaves of *Pst* DC3000 avrRps4 or 10 mM $MgCl_2$ locally inoculated plants at 3 days post-inoculation. **$P < 0.01$ (Student's *t*-test), data are presented as the mean ± SE ($n = 6$).

replicates were considered to be correctly identified and were retained for quantitative analysis. The quantitative analysis was performed on the basis of the LFQ values for the peptides. The following thresholds were used to determine significant increases and decreases in protein abundance: 1.5-fold change and $P < 0.05$ (Student's *t*-test).

The *A. thaliana* TAIR 10 database (arabidopsis.org) was used to functionally annotate the identified proteins. The mass spectrometry proteomics data have been deposited to the ProteomeXchange Consortium (http://proteomecentral.proteomexchange.org) *via* the iProX partner repository (*Ma et al., 2019*).

GO enrichment analysis was performed using Gene Ontology Resource (geneontology. org). The enriched pathways among the identified proteins were determined using the Kyoto Encyclopedia of Genes and Genomes (KEGG) Mapper online tool and then visualized using the Prism (version 7.0) software. The OmicStudio online tools were used for the Venn diagram analysis.

## RESULTS

### Extraction of high-quality AWF

Because SAR is fully established at 48 h after leaves are inoculated with *P. syringae* (*Gruner et al., 2013*), we identified the AWF proteins with significant changes in abundance in the systemic leaves of SAR-induced plants at 48 h. More specifically, *Pst* DC3000 avrRps4 was used to induce SAR in wild-type Col-0 plants. Phenotypic changes and the results of the bacterial growth assay and the RT-qPCR analysis of *PR1* expression confirmed that SAR was successfully induced at 48 h (Fig. 1). The AWF was extracted from the systemic leaves of plants locally inoculated with *Pst* DC3000 avrRps4 or 10 mM $MgCl_2$. The potential contamination of the AWF samples by intracellular compounds was assessed according to electrolyte leakage and malate dehydrogenase (MDH) activity assays. Compared with intact leaves, the conductivity of the leaves used for the AWF extraction increased by less than 1% (0.068%) (Fig. 2A), and the average contamination percentage was 0.76% and

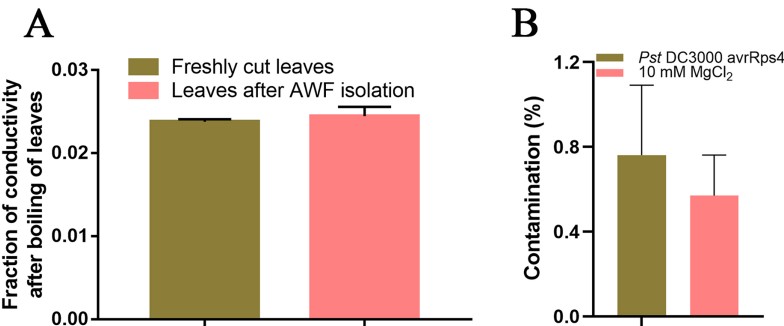

**Figure 2 Intracellular contamination of vacuum infiltrates.** (A) Electrolyte leakage assay results. Leaves (1 g) before and after the AWF extraction were immersed in double-distilled water for 2 h at 25 °C. Electrolyte leakage was measured using a conductivity meter. The total electrolyte leakage from seedlings heated for 2 h at 80 °C was set as 1. The results from three independent experiments are presented as the mean ± SE ($n = 3$). (B) Malate dehydrogenase assay. The results from three independent experiments are presented as the mean ± SE ($n = 3$).     

0.57% for *Pst* DC3000 and 10 mM MgCl$_2$ treated samples (Fig. 2B), respectively. Less than 3% is considered to be negligible contamination according to the previous works (*Alves et al., 2006*; *Zhou et al., 2010*), reflecting the high quality of the AWF, with minimal intracellular component contamination (Fig. 2B).

## Characterization of the AWF proteome

Overall, 747 protein groups with 6,036 peptides were identified with a false discovery rate of 1% (Tables S1 and S2), which represented all the proteins detected in the AWF in systemic leaves of *Pst* DC3000 avrRps4 locally inoculated plants. A variety of the resulting proteins were involved in redox, signal transduction and immune responses (*e.g.*, peroxidases, superoxides, oxidases, kinases, lipases and pathogenesis-related proteins) (Table S2). As shown in Fig. 3A, 705 proteins groups were commonly identified both in the AWF in systemic leaves of *Pst* DC3000 avrRps4 locally inoculated plants and in the AWF in systemic leaves of mock-treated plants (Table S3). GO enrichment analysis was used to further investigate the biological function of proteins identified in SAR- or mock induced plants in this study. For cellular component (CC), the GO terms associated with AWF proteins were mainly enriched in apoplast, cell wall, vacuole, cytosol, plasmodesma, and stromule (Fig. 3B, 3C; Tables S4 and S5). For molecular function (MF), the GO terms associated with AWF proteins were enriched in peroxidase activity, pyridoxal phosphate binding, flavin adenine dinucleotide binding, NAD binding, hydrolase activity, and oxidoreductase activity (Figs. 3B, 3C; Tables S4 and S5). With respect to MF, the GO terms associated with AWF proteins were mainly involved in response to salt stress, response to cytokinin, response to cold, reductive pentose-phosphate cycle, and response to oxidative stress (Figs. 3B, 3C; Tables S4 and S5).

## AWF proteins related to the induction of SAR

To screen for apoplast proteins associated with SAR, AWF protein abundances were compared in systemic leaves of *Pst* DC3000 avrRps4 *vs.* mock (MgCl$_2$) locally inoculated plants. The differentially accumulated proteins (DAPs) were detected on the basis of a 1.5-

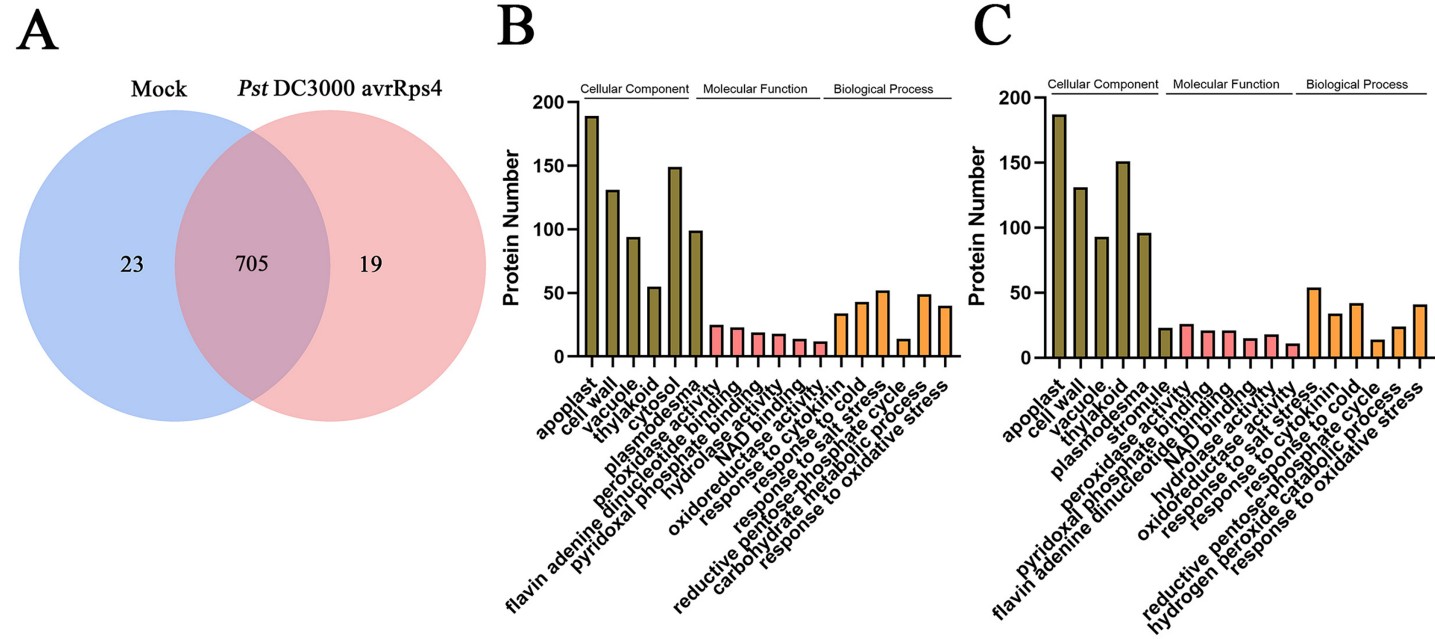

**Figure 3** **Analysis of identified proteins in apoplast washing fluid (AWF).** (A) Comparison of the proteins identified in AWF of *Pst* DC3000 avrRps4- and mock treated plants. (B) GO enrichment analysis of identified proteins in AWF of mock treated plants. (C) GO enrichment analysis of identified proteins in AWF of *Pst* DC3000 avrRps4 treated plants.               

fold difference in abundance and a significance level of $P < 0.05$. Compared with the systemic leaves of the mock control, the abundance of 99 proteins differed in the systemic leaves of the plants inoculated with *Pst* DC3000 avrRps4 (39 upregulated and 60 downregulated). In addition, 10 and eight proteins were detected only in the *Pst* DC3000-infected plants and the mock-treated plants, respectively; these proteins were assigned to the upregulated and downregulated groups, respectively. In total, the abundance of 49 and 68 DAPs respectively increased and decreased in the systemic leaves of SAR-induced plants (Fig. 4A; Table S6). These proteins were assumed to be involved in SAR in the apoplast. The GO and KEGG enrichment analyses were conducted to clarify the functions of these DAPs. In the MF category, these DAPs were mainly associated with the GO terms enzyme activator activity, peroxidase activity, hydrolase activity, oxidoreductase activity, protease binding, polysaccharide binding, and NAD binding (Fig. 4B; Table S7). Considering BP, these DAPs were mainly associated with the GO terms carbohydrate metabolic process, cell wall organization, hydrogen peroxide catabolic process, positive regulation of catalytic activity, response to wounding, and cellular response to oxidative stress (Fig. 4C; Table S7). The KEGG enrichment analysis showed that these DAPs were mainly involved in metabolic pathways, phenylpropanoid biosynthesis, biosynthesis of secondary metabolites, sphingolipid metabolism, glycosaminoglycan degradation, and glycosphingolipid biosynthesis (Fig. 4D; Table S8).

The apoplast serves as one of the first compartments pathogens encounter during the invasion of plant cells. Thus, it can directly inhibit pathogen growth, while also contributing to various crucial plant defense responses, including the strengthening of the

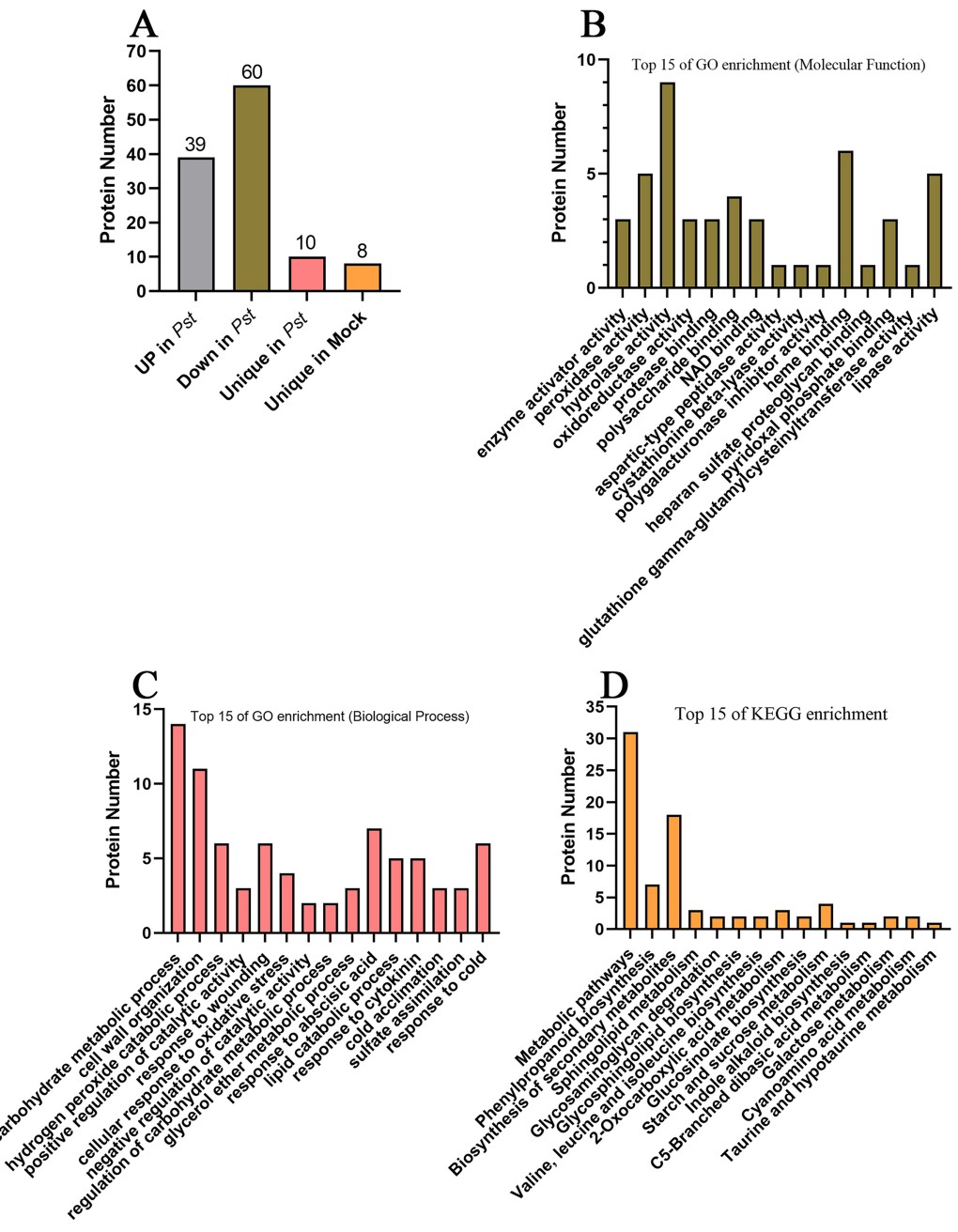

**Figure 4 Analysis of differentially accumulated proteins.** (A) The differentially accumulated proteins identified in this study. (B) Top 15 of GO enrichment on molecular function; (C) top 15 of GO enrichment on biological process; (D) top 15 of KEGG enrichment analysis.

cell wall, the perception and recognition of pathogens, and the transduction of immune signals. Accordingly, the 117 identified DAPs included proteins associated with cell wall modifications, signal transduction, and immune responses. Notably, a series of DAPs, such as AT5G59680 and stress induced factor 3 (SIF3), have the transmembrane domain and are anticipated to be located in the cell membrane. Recent studies have revealed that plants
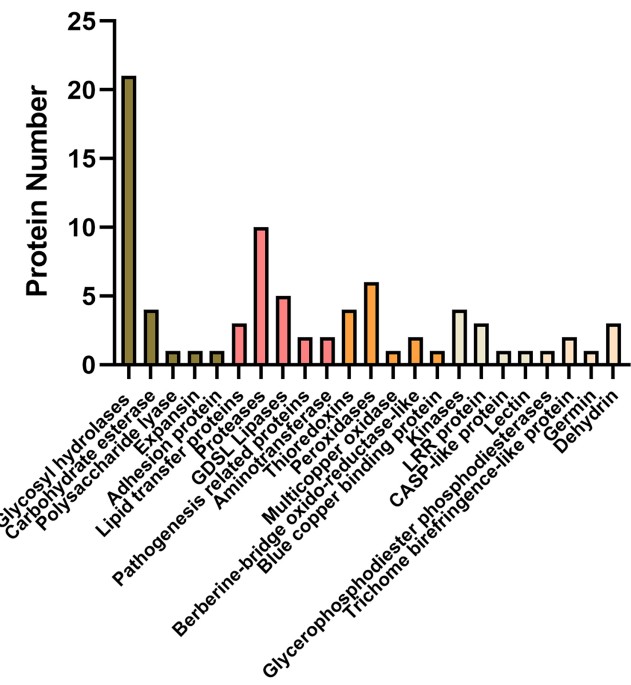

**Figure 5 Representative differentially accumulated proteins in the apoplast washing fluid of the systemic leaves of SAR-induced plants.**

are able to release exosomes, which are small vesicles (30–150 nm) enclosed by a lipid bilayer membrane, into the extracellular space (*Regente et al., 2017*). It has been confirmed that exosomes play essential roles in plant-microbe interactions (*Cai et al., 2018*). Several transmembrane proteins, including the kinase SIF3, have been identified in exosomes of *A. thaliana* (*Rutter & Innes, 2017*). Thus, it's possible that the transmembrane proteins could be secreted to the extracellular space through exosomes.

Of these, a total of 92 proteins were also identified in cell wall proteomics by comparing with *WallProtDB* (http://www.polebio.lrsv.ups-tlse.fr/WallProtDB/) database (*Clemente & Jamet, 2015*; *Clemente et al., 2022*) (Table S9). Specifically, there were twenty-eight DAPs involved in cell wall modifications consisted of twenty-one glycosyl hydrolases (GHs), four carbohydrate esterase, one polysaccharide lyase, one expansin, and one adhesion protein (Fig. 5, Table 1). Additionally, four kinases, namely AT5G59680, cysteine-rich receptor-like kinase 9 (CRK9), SIF3, and nucleoside diphosphate kinase 1 (NDK1), were among the DAPs in the systemic leaves of *Pst* DC3000-inoculated plants, whereas eleven DAPs were related to redox reactions, including six peroxidases, four thioredoxins, and one multicopper oxidase. Numerous DAPs were associated with defense responses, including lipid transfer proteins (LTPs), GDSL lipases, and proteases (Fig. 5; Table 1). These proteins are presumably involved in SAR in the apoplast of systemic leaves.

## Comparative analysis with other proteome data

We also compared the AWF proteome data set with previous proteome data set for the systemic leaves of *Pst* DC3000 avrRpt2-inoculated plants to analyze the changes in protein abundance in the leaves and apoplast (*Kumar et al., 2020*). The abundance of only the

**Table 1 Representative DAPs in the apoplast of the systemic leaves of *Pst* DC3000 avrRps4-inoculated and mock-inoculated plants.**

| Accession number | Gene name | Description | Fold change (*Pst*/CK) |
| --- | --- | --- | --- |
| **Cell wall modification** | | | |
| **Glycosyl hydrolases (GH)** | | | |
| AT5G07830 | GUS2 | Glucuronidase 2 | 1.86 |
| AT5G57550 | XTH25 | Xyloglucan endotransglucosylase/hydrolase 25 | 1.73 |
| AT4G30270 | XTH24 | Xyloglucan endotransglucosylase/hydrolase 24 | 1.66 |
| AT2G06850 | XTH4 | Xyloglucan endotransglucosylase/hydrolase 4 | 0.38 |
| AT5G58480 | / | O-Glycosyl hydrolases family 17 protein | 1.60 |
| AT5G63800 | BGAL6 | Beta-galactosidase 6 | 1.54 |
| AT3G13750 | BGAL1 | Beta galactosidase 1 | 1.52 |
| AT5G56870 | BGAL4 | Beta-galactosidase 4 | 1.51 |
| AT4G26140 | BGAL12 | Beta-galactosidase 12 | 1.51 |
| AT4G36360 | BGAL3 | Beta-galactosidase 3 | Unique in *Pst* |
| AT3G57240 | BG3 | Beta-1,3-glucanase 3 | 1.50 |
| AT1G65590 | HEXO3 | Beta-hexosaminidase 3 | 0.57 |
| AT3G26380 | APSE | Arapase | 1.50 |
| AT2G05790 | / | O-Glycosyl hydrolases family 17 protein | 0.65 |
| AT3G56310 | AGAL3 | Alpha-galactosidase 3 | 0.59 |
| AT5G20950 | BGLC1 | Glycosyl hydrolase family protein | 0.51 |
| AT1G68560 | XYL1 | Alpha-xylosidase 1 | 0.62 |
| AT3G13790 | CWI1 | Cell wall invertase 1 | 0.48 |
| AT4G23820 | PGF13 | Pectin lyase-like superfamily protein | 0.53 |
| AT5G41870 | PGF15 | Polygalacturonase clade F 15 | 1.91 |
| AT3G61490 | PGF9 | Polygalacturonase Clade F 9 | Unique in *Pst* |
| **Carbohydrate esterases** | | | |
| AT2G23610 | MES3 | Methyl esterase 3 | Unique in *Pst* |
| AT3G14310 | PME3 | Pectin methylesterase 3 | 0.46 |
| AT4G19410 | PAE7 | Pectin acetylesterase 7 | 0.56 |
| AT5G23870 | PAE9 | Pectin acetylesterase 9 | 1.62 |
| **Polysaccharide lyase** | | | |
| AT4G24780 | PLL19 | Probable pectate lyase 18 | 0.47 |
| **Expansin** | | | |
| AT1G20190 | EXPA11 | Expansin 11 | 0.56 |
| **Adhesion protein** | | | |
| AT3G46550 | FLA4 | Fasciclin-like arabinogalactan-protein 4 | 1.84 |
| **Defensive** | | | |
| **Lipid transfer proteins (LTP)** | | | |
| AT2G38540 | LTP1 | Lipid transfer protein 1 | 0.34 |
| AT2G45180 | DRN1 | Disease related nonspecific lipid transfer protein 1 | 0.18 |
| AT2G10940 | / | Lipid-transfer protein | 0.11 |
| **Proteases** | | | |
| AT1G15000 | scpl50 | Serine carboxypeptidase-like 50 | 1.65 |

(*Continued*)

| Accession number | Gene name | Description | Fold change (*Pst*/CK) |
|---|---|---|---|
| AT1G03230 | SAP1 | Secreted aspartic protease 1 | 0.52 |
| AT4G34980 | SLP2 | Subtilisin-like serine protease 2 | 0.47 |
| AT4G15100 | scpl30 | Serine carboxypeptidase-like 30 | Unique in *Pst* |
| AT2G33530 | scpl46 | Serine carboxypeptidase-like 46 | 0.29 |
| AT5G07030 | / | Eukaryotic aspartyl protease family protein | 0.17 |
| AT1G66180 | / | Putative aspartyl protease | Unique in CK |
| AT3G18490 | ASPG1 | Aspartic protease in guard cell 1 | 2.58 |
| AT2G24280 | / | Alpha/beta-hydrolases superfamily protein | Unique in CK |
| AT4G21650 | SBT3.13 | Subtilase 3.13 | 0.28 |
| **GDSL lipases** | | | |
| AT1G53920 | GLIP5 | GDSL-motif lipase 5 | 0.60 |
| AT4G01130 | / | GDSL-motif lipase | 0.57 |
| AT3G16370 | GGL19 | GDSL-motif lipase | 0.49 |
| AT5G55050 | / | GDSL-motif lipase | 0.27 |
| AT1G33811 | GGL7 | GDSL-motif lipase | Unique in CK |
| **Pathogenesis related proteins** | | | |
| AT3G12500 | PR3 | Pathogenesis-related 3 | 1.61 |
| AT2G38870 | PR6-like | PR-6 proteinase inhibitor family | 1.57 |
| **Aminotransferase** | | | |
| AT4G23600 | JR2 | Jasmonic acid responsive 2 | 1.85 |
| AT3G19710 | BCAT4 | Branched-chain aminotransferase 4 | Unique in *Pst* |
| **Redox** | | | |
| **Thioredoxins** | | | |
| AT1G50320 | THX | Thioredoxin X | 2.01 |
| AT1G03680 | TRX-M1 | Thioredoxin M-type 1 | 1.61 |
| AT1G21350 | / | Thioredoxin superfamily protein | 1.58 |
| AT3G15360 | TRX-M4 | Thioredoxin M-type 4 | 1.53 |
| **Peroxidases** | | | |
| AT5G19890 | PER59 | Peroxidase 59 | 4.97 |
| AT5G05340 | PRX52 | Peroxidase 52 | 1.95 |
| AT2G37130 | PER21 | Peroxidase 21 | 1.72 |
| AT3G28200 | / | Peroxidase superfamily protein | 0.62 |
| AT5G40150 | / | Peroxidase superfamily protein | Unique in *Pst* |
| AT1G07890 | APX1 | L-ascorbate peroxidase 1 | 0.59 |
| **Multicopper oxidase** | | | |
| AT1G76160 | sks5 | SKU5 similar 5 | 1.50 |
| **Berberine-bridge oxido-reductase-like** | | | |
| AT2G34790 | AtBBE-like 15 | Berberine bridge enzyme-Like 15 | 0.62 |
| AT5G44400 | AtBBE26 | FAD-binding berberine family protein | 0.42 |
| **Blue copper binding protein** | | | |
| AT4G12880 | ENODL19 | Early nodulin-like protein 19 | 0.60 |

| Accession number | Gene name | Description | Fold change (*Pst*/CK) |
|---|---|---|---|
| **Proteins with interaction domains** | | | |
| **Kinases** | | | |
| AT5G59680 | / | Leucine-rich repeat protein kinase family protein | 1.96 |
| AT4G23170 | CRK9 | Cysteine-rich receptor-like kinase 9 | 1.69 |
| AT4G09320 | NDK1 | Nucleoside diphosphate kinase 1 | 0.61 |
| AT1G51805 | SIF3 | Stress induced factor 3 | 1.55 |
| **LRR protein** | | | |
| AT1G49750 | / | Leucine-rich repeat (LRR) family protein | 0.65 |
| AT5G06870 | PGIP2 | Polygalacturonase inhibiting protein 2 | 0.44 |
| AT5G12940 | / | Leucine-rich repeat (LRR) family protein | 0.20 |
| **CASP-like protein** | | | |
| AT5G62360 | PMEI13 | Pectin methyl-esterase inhibitor 13 | 0.49 |
| **Lectin** | | | |
| AT5G03350 | SAI-LLP1 | SA-induced legume lectin-like protein 1 | 0.51 |
| **Miscellaneous** | | | |
| **Glycerophosphodiester phosphodiesterases (GDPD)** | | | |
| AT5G55480 | GDPDL4 | Glycerophosphodiester phosphodiesterase like 4 | Unique in CK |
| **Trichome birefringence-like protein** | | | |
| AT2G34070 | TBL37 | Trichome birefringence-like 37 | 1.56 |
| AT2G37720 | TBL15 | Trichome birefringence-like 15 | 1.52 |
| **Germin** | | | |
| AT1G72610 | GER1 | Germin-like protein 1 | 0.49 |
| **Dehydrin** | | | |
| AT1G20440 | COR47 | Cold-regulated 47 | 0.49 |
| AT1G20450 | LTI45 | Low temperature induced 45 | 0.52 |
| AT1G76180 | ERD4 | Early response to dehydration 14 | 0.41 |

following two proteins increased/decreased similarly in the leaves and apoplast: arginine amidohydrolase 2 (ARGAH2, AT4G08870) and NAD(P)-binding Rossmann-fold superfamily protein (AT2G37660) (Fig. 6A). The abundance of two proteins, including phosphoribulokinase (PRK, AT1G32060) and chaperonin-60beta1 1 (CPN60B, AT1G55490), showed the opposite trend of accumulation in the leaves and apoplast (Fig. 6A; Table S10), suggesting these DAPs were selectively enriched in the apoplast during the induction of SAR.

We further compared our data set with previously published secretory proteome of suspension-cultured cells of *Arabidopsis* infected by *Pst* DC3000 (*Kaffarnik et al., 2009*). The abundance of two proteins, namely ribonuclease 1 (RNS1, AT2G02990), and nucleoside diphosphate kinase 1 (NDK1, AT4G09320), were commonly decreased in *P. syringae* induced secretome of suspension-cultured cells and apoplast of SAR-activated systemic leaves (Fig. 6B; Table S11).
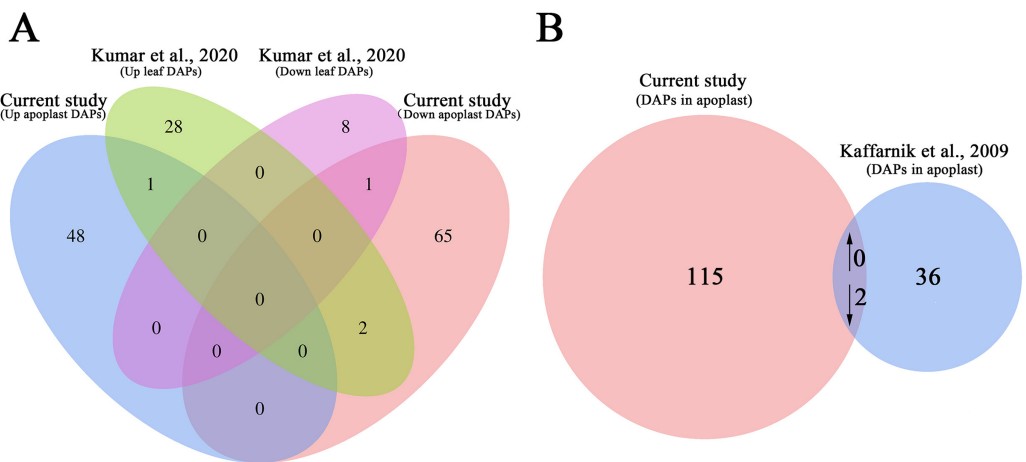

**Figure 6 Comparative analysis of the proteomes from this study and earlier studies.** (A) Comparison of the differentially accumulated proteins identified in this study and in a previous study on the systemic leaf proteome of SAR-induced plants. (B) Comparison of the differentially accumulated proteins identified in this study and in a previous study on the secretory proteome of suspension-cultured cells of *Arabidopsis* infected by *Pst* DC3000.

## DISCUSSION

In *A. thaliana*, the SAR-regulated genes, proteins and metabolites have been identified by numerous works (*Gruner et al., 2013*; *Bernsdorff et al., 2016*; *Kumar et al., 2020*; *Gao et al., 2020*; *Wang et al., 2016a*). In the current study, we first investigated the apoplast proteins related to SAR in the systemic leaves of plants locally inoculated with *Pst* DC3000 avrRps4. A total of 747 protein groups were identified, among which the abundance of 117 proteins, including various peroxidases, PR proteins, kinases, lipases, and oxidases, changed significantly in the apoplast of SAR-induced systemic leaves.

### Obtaining high-quality apoplastic fluid

Vacuum infiltration centrifugation (VIC) is a simple and well-established method for extracting AWF from plant leaves, which has been widely used to to characterize the apoplastic proteome in a variety of plants (*Lohausa et al., 2001*; *Delaunois et al., 2014*; *Petriccionea et al., 2014*). This method allows for the isolation of AWF without causing significant cell damage at low centrifugal forces (<1,000 g) (*Lohausa et al., 2001*). However, the cytoplasmic contamination in AWF can not be completely avoided because of the fragility of the leaf samples (*Delaunois et al., 2014*). Therefore, a more rigorous evaluation of intracellular contamination is needed to ensure the purity of AWF. Electrolyte leakage and malate dehydrogenase activity assays are the most commonly used methods for evaluating the level of cell damage caused by the VIC method (*Delaunois et al., 2014*). Electrolyte leakage assays revealed that the conductivity of the leaves used for the AWF extraction increased by less than 1% (0.068%) compared with intact leaves (Fig. 2A), and the average contamination percentage of MDH was 0.76% and 0.57% for *Pst* DC3000 and 10 mM $MgCl_2$ treated samples (Fig. 2B), respectively, suggesting the AWF isolated by this method was minimally contaminated with intracellular components.

## Cell wall remodeling-related DAPs

Glycoside hydrolases (GHs), which mainly cleave glycosidic bonds between carbohydrates, have essential functions related to cell wall remodeling and chemical defenses (*Minic & Jouanin, 2006*; *Barth & Jander, 2006*). Hemicelluloses (xyloglucan, xylan, and glucomannan) and pectins (galactan and homogalacturonan) are potential substrates for most GHs (*Liu et al., 2021*). ARAPASE (ASPE), which belongs to the GH27 family, helps mediate cell wall remodeling. In *A. thaliana*, a lack of a functional ASPE results in an abnormal cell wall composition (*Imaizumi et al., 2017*). Beta galactosidases (BGALs) are GH family members that catalyze the hydrolysis of terminal β-galactosyl residues, leading to the release of galactose molecules (*Chandrasekar & van der Hoorn, 2016*). *Moneo-Sanchez et al. (2019)* reported that BGAL3 modulates the cell wall architecture by affecting the interactions between cellulose and xyloglucan. Another study showed that xyloglucans, which are hemicellulosic polysaccharides, are the major polymer components in the primary cell wall (*Schultink et al., 2014*). Previous studies have shown that xyloglucans contribute to cell wall structure by cross-linking with cellulose, pectin, and proteoglycans (*i.e.*, arabinogalactan proteins) (*Schultink et al., 2014*; *Tan et al., 2013*). Moreover, xyloglucan endotransglycosylase/hydrolase (XTH; GH16 family) is responsible for integrating newly synthesized xyloglucans into the cell wall and remodeling the pre-existing cell wall xyloglucans through its hydrolase and/or endotransglucosylase activity (*Eklof & Brumer, 2010*; *Rose et al., 2002*). In this study, 15 GHs, including ASPE, BGAL1, BGAL3, BGAL4, BGAL6, BGAL12, XTH4, XTH24, and XTH25, were detected as DAPs in the apoplast of the systemic leaves (Table 1), suggesting that these enzymes may be involved in SAR by remodeling the cell wall (Table 1).

Pectin, comprising mainly of homogalacturonan, rhamnogalacturonan I (RG-I), and the substituted galacturonan rhamnogalacturonan II (RG-II), is important for plant growth, development, and DAMPs-triggered immunity (*Ogawa et al., 2009*; *Hongo et al., 2012*; *Ferrari et al., 2012*; *Lionetti, Cervone & Bellincampi, 2012*). Oligogalacturonides are the most thoroughly characterized DAMPs in plants (*Ferrari et al., 2013*). They are mainly derived from the degradation of a major component of pectin by PGs secreted by pathogens or produced by plants (*Bacete et al., 2018*; *Brutus et al., 2010*; *Benedetti et al., 2018*). The activities of endogenous PGs in plants can promote the accumulation of oligogalacturonides in the extracellular matrix (*Savatin et al., 2014*). *Ohashi et al. (2022)* cloned genes encoding five PGs from *A. thaliana* and observed that these PGs can hydrolyze polygalacturonic acid. In the present study, the PGF9 and PGF15 contents increased significantly in the apoplast of the SAR-induced systemic leaves (Table 1), indicative of a possible role for DTI in the induction of SAR.

Methylesterification and methylesterifications are the most common post-synthesis modifications of pectin (*Atmodjo, Hao & Mohnen, 2013*). The de-methylesterification of pectic polymers by PMEs can generates active oligogalacturonides and methanol, both of which may induce plant immune responses (*Komarova, Sheshukova & Dorokhov, 2014*). In *A. thaliana*, the content of PME17 increases in response to infections by several pathogens and is essential for the resistance against *Botrytis cinerea* mediated through the

jasmonic acid–ethylene-dependent signaling pathway (*Manabe et al., 2011*; *Del et al., 2020*). The de-esterified of oligogalacturonides by PME3 was essential for its function in activating DTI (*Kohorn et al., 2014*). The overexpression of pectin-specific *Aspergillus nidulans* acetylesterase genes in transgenic *A. thaliana* plants significantly decreases cell wall acetylations and increases the resistance to *B. cinerea* (*Pogorelko et al., 2013*). In the current study, three pectin lesterases, including PME3, PAE7 and PAE9, were differentially accumulated in the apoplast of systemic leaves, suggesting their potential contribution to SAR by modifying cell wall pectin (Table 1).

## Defense response-related DAPs

Pathogenesis-related proteins have key functions in plant–microbe interactions. Recent research revealed that PR proteins are the most abundant proteins in plant apoplasts (*Delaunois et al., 2014*). Consistent with this finding, several PR proteins, including PR1, PR3, PR5, LTPs, and several GDSL-motif lipases, were identified in this study (Table S2). Lipid transfer proteins, which belong to the PR14 family, have diverse functions in plant immune responses (*Gao et al., 2022b*). *Dhar et al. (2020)* reported that DRN1 expression is considerably downregulated after a pathogen infection. A functional DRN1 is necessary for defense responses to biotic factors (*e.g.*, pathogenic fungi and bacteria) as well as abiotic stresses (*Dhar et al., 2020*). In *A. thaliana*, LTP1 was demonstrated to interact directly with REVERSION-TO-ETHYLENE SENSITIVITY1 and perform a regulatory role in ethylene receptor signaling (*Wang et al., 2016b*). Three LTPs (LTP1, DRN1, and AT2G10940) were identified as DAPs in the apoplast of the systemic leaves of SAR-induced plants, implying these LTPs participate in the establishment of SAR in the apoplast (Table 1). Lipases with a GDSL-like motif are considered to function like PR proteins (*Jakab et al., 2003*).

For example, GDSL lipases, which are a subfamily of lipolytic enzymes characterized by a conserved GDSL motif, influence plant defense responses by enhancing the biosynthesis of the natural insecticide pyrethrin and promoting glucosinolate metabolism (*Gao et al., 2022a*). We detected five GDSL lipases that accumulated differentially in the apoplast (Table 1), indicating their potential involvement in SAR.

In addition to PR proteins, several other plant immune response-related DAPs were identified, including proteolytic enzymes, thioredoxins, kinases, and peroxidases. The potential roles of thioredoxins, kinases, and peroxidases will be discussed later. Jasmonates (JAs) are phytohormones that affect plant responses to various biotic stresses. *Yang et al. (2008)* found that methyl esterase 3 (MES3) catalyzes the hydrolysis of methyl jasmonate (MeJA) to produce JA. Though JA levels are unaffected in the systemic leaves of SAR-induced plants, JA signaling is likely not involved in the induction of SAR (*Gruner et al., 2013*), However, the accumulation of MeJA in the systemic guard cells of SAR-induced plants suggests that JA may be associated with the SAR in certain cells (*David, Kang & Chen, 2020*). This possibility is supported by our observation that MES3 and jasmonic acid-responsive 2 (JR2) were enriched in the apoplast of systemic leaves (Table 1).

## Redox-related DAPs

The activation of PTI and ETI is associated with the rapid production of large amounts of ROS, which dramatically disrupt cellular redox homeostasis (*Mata-Perez & Spoel, 2019*). The accumulated ROS contribute to plant immune responses in a variety of ways (*e.g.*, increasing cell wall rigidity and strength, inducing PCD, and activating immune signaling pathways) (*Mata-Perez & Spoel, 2019*). In addition to their effects on local resistance, ROS are also important for the induction of SAR in bacteria-free systemic tissues (*Wang et al., 2014*). However, the excessive accumulation of ROS can cause widespread cell damages. Thus, plants have evolved multiple antioxidant scavengers of these highly reactive molecules, including peroxidases, glutaredoxins, and many other reductases (*Mittler et al., 2011*). In the current study, the accumulation of almost all of the redox-related DAPs, including four thioredoxins (THX, TRXm1, TRXm4, and AT1G21350) and five peroxidases ( PER21, PRX52, PER59, AT3G28200, and AT5G40150), increased in the systemic apoplast of the SAR-induced plants (Table 1), which may reflect the roles of ROS during the induction of SAR.

In addition to regulating ROS homeostasis, the above-mentioned redox-related DAPs also help regulate plant immune signaling pathways. The absence of functional TRXm1, or TRXm4 has detrimental effects on SAR (*Carella et al., 2016*). However, it remains unclear how these TRXs function in the systemic apoplast during the induction of SAR. In *A. thaliana*, TRXh5 and TRXh3 contribute to SA-induced immunity-related gene expression by regulating the cytosolic NPR1 conformation. Both TRXh5 and TRXh3 catalyze the reduction of the disulfide bonds in the NPR1 oligomer, thereby releasing the active monomer of NPR1 to induce the expression of SA-responsive genes (*Tada et al., 2008*). *Manohar et al. (2014)* reported that TRXm1 can bind to the defense hormone SA. Considering their ability to reduce cysteine, TRXm1/4 and other differentially accumulated TRXs may be involved in the redox regulation of target metabolites and proteins during the induction of SAR. This possibility is currently being experimentally verified as part of our ongoing research.

Five peroxidases were identified as DAPs in the systemic apoplast of SAR-induced plants. Except for AT3G28200, the abundances of these peroxidases increased (Table 1). In addition to modulating ROS homeostasis, peroxidases also influence plant immunity *via* other mechanisms. For example, PRX52 affects the lignin composition of secondary cell walls (*Fernandez-Perez et al., 2015*). *Smith et al. (2021)* reported that PER52 may serve as an ATP receptor in the extracellular space and may be involved in ATP-mediated stress-adaptive processes. The genes encoding the differentially accumulated peroxidases in the systemic apoplast may be useful for future investigations on the mechanism underlying the induction of SAR.

## Other DAPs

Many other proteins were differentially accumulated in the systemic apoplast of SAR-activated plants, including kinases (AT5G59680, SIF3, CRK9, and NDK1),

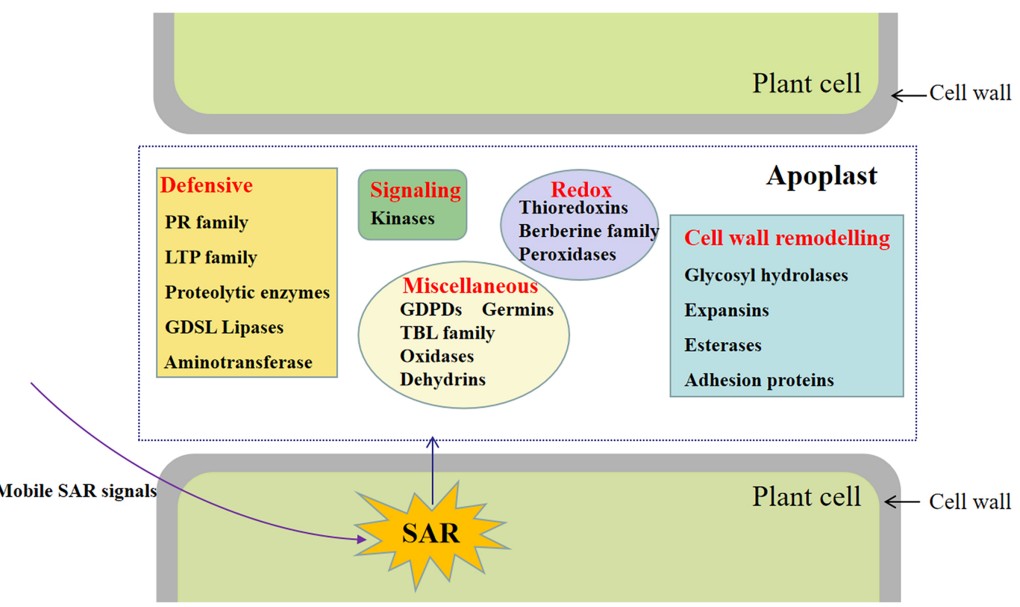

**Figure 7 Schematic overview of DAPs involved in SAR induction in apoplast of systemic leaves in SAR-induced plants.** The diagram shows representative proteins.

phosphodiesterases (*e.g.*, GDPDL4), oxidases (SKS5), trichome birefringence-like family members (TBL15 and TBL37), and berberine bridge family members (BBE26 and BBE-like 15) (Table 1). The nucleoside diphosphate kinase NDK1 mediated *A. thaliana* ROS signaling by interacting with catalases. *NDK1* overexpression plants had higher ability to eliminate $H_2O_2$ than wild type plants (*Fukamatsu, Yabe & Hasunuma, 2003*).

The germin-like protein positively regulates plant responses to salt stress (*Chen et al., 2021*). *Daniel et al. (2015)* reported that the berberine bridge enzyme BBE-like 15 plays a role in plant cell wall metabolism by oxidizing monolignin. Furthermore, GDPDL4 and its paralog SHV3 affect the primary cell wall organization by altering the cellulose content and pectin modifications (*Hayashi et al., 2008*). The functions of these proteins related to plant development and abiotic stress responses have been well documented. However, the potential contributions of these proteins to plant SAR will need be investigated and verified.

## CONCLUSIONS

This study involved the analysis of the SAR-specific proteome in apoplasts of systemic leaves of SAR-induced and mock-treated plants. A total of 747 protein groups were identified, among which 117 were identified as DAPs in the apoplast of the systemic leaves of SAR-induced plants (*e.g.*, kinases, proteins related to cell wall modifications, defense response-related proteins, and proteins associated with redox reactions) (Fig. 7). To the best of our knowledge, this is the first study to focus on the SAR-related proteins in the systemic apoplast.

### Funding

This work was supported by the Key Scientific Research Project of Colleges and Universities in Henan Province (No. 23B210004), and the Natural Science Foundation of Henan Province (222300420259). The funders had no role in study design, data collection and analysis, decision to publish, or preparation of the manuscript.

### Grant Disclosures

The following grant information was disclosed by the authors:
The Key Scientific Research Project of Colleges and Universities in Henan Province: 23B210004.
Natural Science Foundation of Henan Province: 222300420259.

### Competing Interests

The authors declare that they have no competing interests.

### Author Contributions

- Shuna Jiang conceived and designed the experiments, performed the experiments, analyzed the data, prepared figures and/or tables, and approved the final draft.
- Liying Pan performed the experiments, analyzed the data, prepared figures and/or tables, and approved the final draft.
- Qingfeng Zhou analyzed the data, prepared figures and/or tables, and approved the final draft.
- Wenjie Xu analyzed the data, prepared figures and/or tables, and approved the final draft.
- Fuge He analyzed the data, prepared figures and/or tables, and approved the final draft.
- Lei Zhang conceived and designed the experiments, authored or reviewed drafts of the article, and approved the final draft.
- Hang Gao conceived and designed the experiments, authored or reviewed drafts of the article, and approved the final draft.

### Data Availability

The mass spectrometry proteomics data are available at the ProteomeXchange Consortium *via* the iProX partner repository: PXD038737. https://www.iprox.cn/page/subproject.html?id=IPX0005584002.

The raw measurements are available in the Supplemental Files.

### Supplemental Information

Supplemental information for this article can be found online at http://dx.doi.org/10.7717/peerj.16324#supplemental-information.

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
