# Peer review of "Analysis of the apoplast fluid proteome during the induction of systemic acquired resistance in Arabidopsis thaliana"

_PeerJ, doi:10.7717/peerj.16324_

## Round 0.1 · original submission · Major Revisions

Your manuscript has been reviewed by three reviewers and all of them are positive about your efforts in describing the apoplast proteome in systemic leaves. Please go carefully through the numerous valuable suggestions and consider even the suggestion to shorten the discussion and to move some of the sections into the introduction. In addition, please pay careful attention to the advice of the reviewer about the validity of your findings (reviewer #3) and potential contaminations (reviewer #1).

·

Basic reporting

Overall, this is a nice and interesting paper on the dificult and underexplored subject of the proteins in the apoplast. Overall the background and results are clearly reported. I have a few suggestions for improvements.

The paper would be improved by substantially shortening the discussion, going into all detail about every subset of putative apoplastic proteins found is not helpfull to the reader. I would suggest a substantial shortening of this section.

While I like the effort to compare to other proteomics studies, it seems it would be very easy to make some comaprisions as well to publically available expression data for the identified proteins.

I believe there should be a little clearer more formal defintion of SAR presented in the instroduction.

In figure 1, clarify that it's growth of DC3000 in systemic leaves that is shown in panel B and C.

Figure 2 seems a little bit superflous as it is now. If it is kept, clarify that the "unit" is something like "Fraction of conductivity after boiling of leaves".

A final language editing would also improve the manuscript.

Experimental design

The experimental design is straight forward and seems robust enough. The one thing I'm missing is a little more carefull dscussion and analysis on how well the method will select for apoplastic proteins, surely there will be some contamination of intracellular proteins.

In the material methods section I'm missing information about the light conditions for plant growth as well as which wild type was used.

Please add informaiton about the compostion of the mobile phases used for the proteomics LC.

Validity of the findings

The findings are valid allthough some of the conclusions might be a little stretched, shortening the discussion would improve this.

Reviewer 2 ·

Basic reporting

In general, the manuscript is well written. Some important background information are provided in ‘Results’ or ‘Discussion’. I strongly suggest to described these in ‘Introduction’ for easier reading. Regarding MS data, more information needed to be deposited to the repository.

1. Line 50: Since most plants are resistant against potentially pathogenic microbes and most of pathogens have rather narrow host ranges, I’m not sure whether the description ‘some plants can recognize…’ is accurate. The authors may consider to rephrase.
2. Line 57: How about against ‘oomycetes’?
3. Line 100: If I’m correct the apoplast (but probably not apoplastic proteome) is known to be important for “the induction” of SAR. The authors may want to say “how or whether the induction of SAR affect the apoplastic proteome is not yet known”, which is the aim of this study. Am I wrong?
4. The study by Carella et al. 2016, which is highly relevant to this study, should be introduced in ‘Introduction’. As the authors described in ‘Result’, Carella et al. analyzed phloem exudates collected from local leaves upon the induction of systemic acquired resistance (SAR) triggered by Pst DC3000 avrRpt2 inoculation in Arabidopsis thaliana. The authors should describe major differences between the studies, which would highlight the significance of this study.
5. The authors may consider to cite the following articles as well.
Petriccione et al., 2014, https://doi.org/10.1016/j.jprot.2014.01.030
Kaffarnik et al., 2009, https://doi.org/10.1074/mcp.M800043-MCP200
Cheng et al., 2009, https://doi.org/10.1021/pr800649s
6. Line 110: A. thaliana in Italic.
7. Regarding MS data, Metadata and MaxQuant out put files should also be submitted to public repository, which I failed to find at the iProX. These information/data are indispensable for the evaluation and to utilize the submitted rawdata by readers.
8. Line 201 ‘The identification of this many proteins…’: This information is not needed and not informative. The MS used in this study is good one but not the high-end one (I would say standard one in these day), and the number of ID is normal in my opinion. The authors may just delete this sentence.
9. Line 206 ‘the AWF proteins were mainly enriched…’: This description is wrong or misleading. The AWF proteins are not enriched in these compartments. Specific GO terms associated with the AWF proteins are enriched. The authors need to revise the description. Same applies to other related/similar descriptions in the manuscript.
10. Line 236: ‘growth and/or colonization’ instead of ‘infections’?
11. Line 252: I was not sure if Gao, 2021 (dissertation) is published and publicly available. If ‘yes’, it would be good to provide a link for access. If ‘not’, unpublished information cannot be used for the manuscript. From the name, I guess the dissertation is of the last author of this manuscript.

Experimental design

The manuscript by Jiang et al. investigated how apoplastic proteome of systemic leaves is affected upon the induction of systemic acquired resistance (SAR) using an Arabidopsis thaliana - Pst DC3000 avrRps4 model system. As far as I know, as the authors stated, there is no study described on apoplastic proteome of systemic leaves during SAR induction using the recent proteomics approach, and thus this study would be a useful resource for understanding SAR. Performed experiments are reasonably designed. ‘Materials & Methods’ requires more detailed information.

1. Line 110: What was used for growing plants, soil or medium? What was a light source? Please provide more detailed information.
2. How many leaves per plant were infiltrated? How many systemic leaves were collected per plant? Single leaf was infiltrated and single systemic leaf was collected (120 leaves from 120 A. thaliana plants?)? What was the definition of biological replicates here? 120 plants for one replicate?
3. How leaves were infiltrated in a 200 mL needleless syringe? Vacuum applied? How infiltration was confirmed? By the change of color?
4. Line 190 & Figure 1B,C: At which DPI(day post infection)? Which test was applied for statistics? How many replicates were analyzed? Please provide information to ‘Materials & Methods’ and a figure legend.
5. Line 199: How number of proteins was calculated from MaxQuant result? 985 ‘protein groups’ instead of 985 ‘proteins’?
6. Please provide information on how many (unique) peptides were identified and used for quantification for each protein group to supplementary table?
7. Regarding MaxQuant search, how 6 rawdata processed? I guess that 6 rawdata were analyzed together. How data were normalized and did the authors applied match-between-run (MBR)? For DAP analysis which values were used? The authors should use LFQ values for the purpose, but it was not clear from the description and supplementary table. Moreover, how mussing values were treated? Did the authors imputed values? Please provide more detailed information how MS data were processed, which is a minimum requirement for publishing MS data.
8. Figure 3A, Figure 5: It was not clear how proteins were classified/annotated. Please provide information and/or the annotation file used for the analysis. There are many proteins classified as ‘others’. How ‘the representative proteins’ were defined? Is there any statistical support? Or, are these cherry picking by the authors?
9. Figure 3: Analysis of combined data is less informative. I suggest to analyze AWF proteomes of mock treated and SAR-induced conditions separately.
10. Line 217: I suggest to provide the information on proteome overlaps as a venn diagram in Figure 3 or 4.

Validity of the findings

This study provides evidence that the apoplastic proteome is involved in SAR. However, I’m not sure whether it is appropriate to conclude that “the apoplast is involved in SAR induction’. There is no result supporting ‘induction’. As described in other section, number of replicates and statistical methods used for analyses should be provided for all experiments.

1. Line 242: Are these kinases predicted to be secreted? If these are transmembrane receptor-like kinases, why these were identified? The authors may briefly discuss about it.

Reviewer 3 ·

Basic reporting

The Introduction and Discussion are both very selective in reference to the identities and number of SAR regulators identified. They are neither comprehensive nor lacking detail as to why only specific SAR inducers have been referenced. For instance, if the intent in the Introduction is to only refer to mobile SAR inducers (which in my opinion would not be accurate any way), then salicylic acid is missing. SA mobility was shown to be essential for SAR recently. That said, it is essential to acknowledge other known SAR inducers like pinenes, eNADP, NO, ROS, etc. The discussion has the same issue- first two lines of discussion (269-270) are downright inaccurate. Several other chemicals accumulate very quickly and are known to alter gene expression.

Experimental design

This is adequate

Validity of the findings

The results section is missing the most crucial information of all- the identity of the so called 78 up-regulated and 4 downregulated proteins. Either specifically point these out in supplemental table 2 or put that information in the main figures (preferred).
Most importantly, there is no validation of the proteomics work done here. First, there needs to be validation (beyond electrolyte leakage) to show that AWF fractions is truly free of cytoplasmic proteins. A simple staining for RUBISCO or preferably a Western blot for a soluble protein. Protein blot validation for presence of at least one of the most abundant AWF proteins is also needed. As such, the information in Fig 7 is entirely unsubstantiated.

Additional comments

This is an interesting study which would be highly informative to the field IF meaningful data were presented. The current format for data presentation provides the reader with no meaningful understanding of what changes are happening in the SAR activated AWF. Therefore, I disagree with the authors conclusion that "the presented findings provide the basis for future research on....." (lines 470-471)

---

## Round 0.2 · Minor Revisions

Both reviewers acknowledge your revisions and I would like to ask you to check once more your conclusions section as has been suggested by reviewer #2.

·

Basic reporting

I believe the authors have made significant improvements and have met mine and the other reviewers points in a careful manner.

Experimental design

The experimental design is adequate and me points about intracellular contamination has been met. More information about the methods has been provided to meet both mine and the other reviewers concerns.

Validity of the findings

Findings appear well founded.

Reviewer 2 ·

Basic reporting

The authors reasonably responded to comments by all reviewers.

In 'Conclusions' section, I think '985 proteins' supposed to be '747 protein groups'.
I have no further comment other than this.

Experimental design

no comment

Validity of the findings

no comment

Additional comments

no comment

---

## Round 0.3 · Minor Revisions

Both reviewers acknowledge your revisions and I would like to ask you to check once more your conclusions section as has been suggested by reviewer #2. In addition, the GO and KEGG annotations exist solely as string descriptions and have no links to the encoding, especially those of the GO (e.g. GO:12345). I would like to ask you therefore to include them within the supplemental tables as there is no sense of which sequence IDs can be assigned to the categories summarized (maybe Up/Down in S3, but no GO term). Likewise these are solely categorized in the histograms and Venn diagrams without being able to assign which sequence IDs belong to which grouping. Refinement of this data would greatly improve the following of the observations presented within the manuscript.

---

## Round 0.4 · accepted · Accept

Thank you very much for addressing all the suggestions that have been made.